## PERSPECTIVE

# When the brain listens to the kidney, is it TRP'n or what?

John W. Osborn[1] , Louise C. Evans[1], Christopher T. Banek[1] ,
Lucy Vulchanova[2] and Alex Dayton[3]

[1]*Department of Surgery, Division of Autonomic Neuromodulation, University of Minnesota, Minneapolis, MN, USA*
[2]*Department of Neuroscience, Minneapolis, MN, USA*
[3]*Department of Medicine, Division of Nephrology, Minneapolis, MN, USA*

Email: osbor003@umn.edu

Handling Editors: Vaughan Macefield & Diana Martinez

The peer review history is available in the Supporting Information section of this article (https://doi.org/10.1113/JP290333#support-information-section).

Interoception, referring to how the brain senses and processes the internal state of the body, is critical to homeostasis (Khalsa et al., 2018). It is rapidly reemerging as an important field of study due to its broad impact on human health. Sherrington first introduced the concept of interoception when he coined the terms 'interoceptor', and 'interoception' (Khalsa et al., 2018). Pavlov later proposed that changes in the 'milieu interieur' could serve as a conditioned stimulus, which led to his concept of 'interoceptive conditioning' (Khalsa et al., 2018). In this issue of the *Journal of Physiology*, Sullivan and colleagues present an elegant approach to interoception in the kidney using the TRPV1$^{-/-}$ rat (Sullivan et al., 2025).

Currently there is great interest in the interoceptive function of the kidney, largely due to results of clinical trials for catheter-based renal denervation (CBRDN) for the treatment of hypertension. The original concept was that CBRDN decreased blood pressure as a result of ablation of renal *efferent* nerves that regulate kidney function; however, the kidney is also innervated by afferent (sensory) nerves, which project to the central nervous system to modulate global autonomic activity and neurohumoral regulation. In addition to decreasing blood pressure, CBRDN has resulted in decreased incidence of arrhythmias, reductions in muscle sympathetic nerve activity and improved glucose metabolism (Evans et al., 2025). This led to the hypothesis that the benefits of CBRDN are secondary to the ablation of *renal afferent nerves*. In other words the efficacy of CBRDN is, at least in part, the result of modulation of renal interoceptive pathways.

Our group developed a method for targeted afferent renal denervation (ARDN) to test this hypothesis in preclinical models (Foss et al., 2015). Briefly periaxonal application of the TRPV1 agonist, capsaicin, to the renal nerves results in ablation of renal afferent, but not efferent nerves. This method is based on the idea that renal afferent nerves express the TRPV1 channel, but renal efferent nerves do not. Using this approach, our group and others found that ARDN is as effective as total RDN (efferent + afferent: TRDN) at decreasing blood pressure in multiple models of hypertension associated with increased sympathetic nerve activity (Evans et al., 2025). Taken together these studies raise important questions, which have yet to be fully answered, regarding the mechanisms underlying renal interoception. Do all renal afferent nerves express TRPV1 channels, making them sensitive to capsaicin-induced ablation? If not, what percentage do express TRPV1 channels, and what are their sensory modalities? Similarly, are there anatomical differences regarding the presence or absence of TRPV1-expressing afferent renal nerves? Answering these questions is important to advancing the field of renal interoception.

Sullivan and colleagues directly address these questions in a series of experiments in which afferent renal nerve activity (ARNA) was directly measured in response to two distinctly different stimuli: a reduction in renal blood flow or an increase in renal pelvic pressure (Sullivan et al., 2025). Experiments were conducted in a newly developed Trpv1$^{-/-}$ rat model and wild-type (WT) controls. There were two key findings in the study. First total occlusion of the renal artery for 90s resulted in a biphasic increase in ARNA in WT rats with an abrupt increase in the first 45s followed by a delayed further increase beginning at 45s of occlusion. Whereas the first phase was similar in Trpv1$^{-/-}$ and WT rats, the second phase was absent in Trpv1$^{-/-}$ rats. Interestingly partial occlusion of the renal artery resulted in flow-induced increases in ARNA that were similar in Trpv1$^{-/-}$ and WT rats. Second the increase in ARNA induced by increased renal pelvic pressure, a well-known stimulus of afferent renal nerves, was similar in Trpv1$^{-/-}$ and WT rats, suggesting this stimulus for activation of ARNA is not dependent on TRPV1 channels in renal sensory nerves. The authors conclude that TRPV1$^{+}$ channels play a role in chemosensitive but not mechanosensitive stimuli.

This study is significant both for its novel findings and the questions it raises. It strongly suggests that TRPV1 channels play distinct roles in renal interoception and that not all renal sensory nerves are TRPV1$^{+}$. This is important in that the method for 'ARDN' is based on targeting TRVP1 channels with capsaicin and therefore may not result in 100% afferent renal denervation. Interestingly ARDN is confirmed by the absence of the sensory neurotransmitter calcitonin gene-related peptide (CCRP). Does this mean that renal sensory neurons that do not express TRPV1 do not utilise CGRP? Also, the percentage of afferent renal nerves that are TRPV1$^{+}$ is not clear but estimated by Stocker and Sullivan (Stocker & Sullivan, 2023) to be 85–90% using immunofluorescence. Differences in this expression are important to understand as 'ARDN' by capsaicin is effective in reducing blood pressure in some, but not all, preclinical models of hypertension. Finally another important finding in this study is that intrapelvic administration of the TRPV1 antagonist capsazepine reduced the ARNA response to increased pelvic pressure not only in WT but also Trpv1$^{-/-}$ rats. This suggests that prior reports in which this approach was used, indicating that the ARNA response to increased pelvic pressure was mediated by TRVP1 channels in renal sensory nerves, were in fact the result of non-specific effects of this drug.

In summary the study of Sullivan, DeLalio and Stocker provides new clues into our understanding of how the kidney 'talks' to the brain and the role of TRPV1$^{+}$ nerves in this communication. Not only does it provide important new evidence, but it also raises new questions that must be answered to move the field forward.

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

## Additional information

### Competing interests

No competing interests declared

### Author contributions

All the authors contributed to the conception or design of the work; drafting the work or revising it critically for important intellectual content; the final approval of the version to be published; and the agreement to be accountable for all aspects of the work. All the authors have approved the final version of the manuscript and agreed to be accountable for all aspects of the work. All persons designated as authors qualify for authorship, and all those who qualify for authorship are listed.

### Funding

None.

### Keywords

interoception, renal afferent nerves, TRPV1

### Supporting information

Additional supporting information can be found online in the Supporting Information section at the end of the HTML view of the article. Supporting information files available:

**Peer Review History**

