## [Peer Review History · The Journal of Physiology]

"When the brain listens to the kidney, is it TRP'n or what?"

John W Osborn, Louise C. Evans, Christopher T Banek, Lucy Vulchanova, and Alex Dayton

DOI: 10.1113/JP290333

Corresponding author(s): John Osborn (osbor003@umn.edu)

The following individual(s) involved in review of this submission have agreed to reveal their identity: Sean D Stocker (Referee #1)

Review Timeline:

Submission Date:	31-Oct-2025
Editorial Decision:	10-Nov-2025
Revision Received:	26-Nov-2025
Editorial Decision:	08-Dec-2025
Revision Received:	08-Dec-2025
Editorial Decision:	10-Dec-2025
Revision Received:	10-Dec-2025
Accepted:	11-Dec-2025

Senior Editor: Vaughan Macefield

Reviewing Editor: Diana Martinez

Transaction Report:

Re: JP-P-2025-290333 ""When the brain listens to the kidney, is it TRP'n or what?"" by John W Osborn, Louise Evans, Christopher T Banek, Lucy Vulchanova, and Alex Dayton

Dear Dr Osborn,

Thank you for submitting your manuscript to The Journal of Physiology. It has been assessed by a Reviewing Editor and by 1 expert referee and you are now invited to respond to the review comments and submit a revised version for further consideration.

The review comments are copied at the end of this email.

Please address all the points raised and incorporate all requested revisions or explain in your Response to Referees why a change has not been made. We hope you will find the comments helpful and that you will be able to return your revised manuscript within 4 weeks. If you require longer than this, please contact journal staff: jp@physoc.org. Please note that this letter does not constitute a guarantee for acceptance of your revised manuscript.

REVISION CHECKLIST:

IMPORTANT POINTS TO NOTE WHEN REVISING YOUR MANUSCRIPT:

We look forward to receiving your revised submission.
If you have any queries, please reply to this email and we will be pleased to advise.

Yours sincerely,

Vaughan Macefield
Senior Editor
The Journal of Physiology

EDITOR COMMENTS

Reviewing Editor:

This is a perspective article on a accepted manuscript by Sullivan et al on TRPV1 channels. This has been reviewed by a reviewer and the reviewing editor. There are some issues that need to be addressed. Specifically, there seems to be some wording issues on the findings of the original article. Sullivan et al tested whether TRPV1 channels mediate the ARNA responses. This is one of the major comments made by the reviewer, TRPV1 channels versus TRPV1 fibers. The reviewing editor enjoyed the title.

Senior Editor:

Thank you for submitting your invited Perspectives article to The Journal of Physiology. As you will see, the authors of the original article require clarification as to whic TRPV channels you are referring to: I believe you can address this easily and look forward to receiving your revised version shortly.

REFEREE COMMENTS

Referee #1:

Thank you for writing a prospective article on our recently published findings. I do have a few comments below.

1. The perspective article repeated states that TRPV1+ neurons do not mediate ARNA responses to pelvic pressure or mechanosensitive stimuli (lines 71-72, lines 74-75, lines 86-87, lines 89-90). We believe this statement is inaccurate. Our findings tested whether the TRPV1 channel does.

Thus, a clear distinction should be made between TRPV1 fibers versus the TRPV1 channel. Our study did not test whether TRPV1+ versus TRPV1- "fibers" mediate the ARNA responses to renal occlusion or pelvic pressure. Instead, we tested whether the TRPV1 channel mediates the response using a Trpv1-/- rat. Thus, the channel is deleted but the sensory fiber is still present and expressing other channels and receptors. Therefore, the ARNA responses to pelvic pressure or partial reduction in blood flow may be sensed by either TRPV1-positive or non-TRPV1 fibers via a mechanism independent of TRPV1 channel. Note, the Trpv1-/- rat is very different from other approaches such as periaxonal application of capsaicin which distinguish between TRPV1-expressing fibers versus non-TRPV1 fibers. Our study did not test whether TRPV1+ versus TRPV1- fibers mediate responses. Therefore, my suggestion would perhaps use the word "channel" instead of "fiber" or rephrase sentences that denoted "TRPV1+ neuron".

2. Lines 80. The reference provided is incorrect. I believe the authors may be referring to Stocker and Sullivan (Hypertension 2023). The sentence should also indicate this study was performed using immunofluorescence (as electrophysiological methods indicate a lower proportion of TRPV1 renal sensory neurons ~70% - Wang et al AJP Reg 2010) and in rats as there are likely species differences in TRPV1 expression. Each technique has advantages and disadvantages.

END OF COMMENTS

Reviewing Editor:

This is a perspective article on an accepted manuscript by Sullivan et al on TRPV1 channels. This has been reviewed by a reviewer and the reviewing editor. There are some issues that need to be addressed. Specifically, there seems to be some wording issues on the findings of the original article. Sullivan et al tested whether TRPV1 channels mediate the ARNA responses. This is one of the major comments made by the reviewer, TRPV1 **channels** versus TRPV1 **fibers**. The reviewing editor enjoyed the title.

Response: Thank you. We have responded to these comments by the reviewer as shown below.

Senior Editor:

Thank you for submitting your invited Perspectives article to The Journal of Physiology. As you will see, the authors of the original article require clarification as to which TRPV channels you are referring to: I believe you can address this easily and look forward to receiving your revised version shortly.

Response: Thank you. We have responded to these comments by the reviewer as shown below.

Referee 1:

Thank you for writing a prospective article on our recently published findings. I do have a few comments below.

1. The perspective article repeated states that TRPV1+ neurons do not mediate ARNA responses to pelvic pressure or mechanosensitive stimuli (lines 71-72, lines 74-75, lines 86-87, lines 89-90). We believe this statement is inaccurate. Our findings tested whether the TRPV1 channel does.

Thus, a clear distinction should be made between TRPV1 fibers versus the TRPV1 channel. Our study did not test whether TRPV1+ versus TRPV1- "fibers" mediate the ARNA responses to renal occlusion or pelvic pressure. Instead, we tested whether the TRPV1 channel mediates the response using a *Trpv1^{-/-}* rat. *Thus, the channel is deleted but the sensory fiber is still present and expressing other channels and receptors.* Therefore, the ARNA responses to pelvic pressure or partial reduction in blood flow may be sensed by either TRPV1-positive or non-TRPV1 fibers via a mechanism independent of TRPV1 channel. *Note, the *Trpv1^{-/-}* rat is very different from other approaches such as periaxonal application of capsaicin which distinguish between TRPV1-expressing fibers versus non-TRPV1 fibers.* Our study did not test whether TRPV1+ versus TRPV1- fibers mediate responses. Therefore, my suggestion would perhaps use the word "channel" instead of "fiber" or rephrase sentences that denoted "TRPV1+ neuron".

Response: Thank you for raising this subtle but very important issue regarding terminology regarding TRVP1. We agree this is critical distinction which we have addressed in the revised Perspectives. As suggested, we have replaced the words “nerves or fibers” with “channels” where appropriate as indicated in the revised text.

2. Lines 80. The reference provided is incorrect. I believe the authors may be referring to Stocker and Sullivan (Hypertension 2023). The sentence should also indicate this study was performed using immunofluorescence (as electrophysiological methods indicate a lower proportion of TRPV1 renal sensory neurons ~70% - Wang et al AJP Reg 2010) and in rats as there are likely species differences in TRPV1 expression. Each technique has advantages and disadvantages.

Response: You are correct, and we apologize for this oversight. The reference has been corrected, and we have modified the sentence to indicate that the estimation of 85-90% was based on using immunofluorescence.

Dear Dr Osborn,

Re: JP-P-2025-290333R1 ""When the brain listens to the kidney, is it TRP'n or what?"" by John W Osborn, Louise C. Evans, Christopher T Banek, Lucy Vulchanova, and Alex Dayton

Thank you for submitting your manuscript to The Journal of Physiology. It has been assessed by a Reviewing Editor and by 1 expert referee and we are pleased to tell you that it is acceptable for publication following satisfactory minor revision.

The review comments are copied at the end of this email.

Please address all the points raised and incorporate all requested revisions or explain in your Response to Referees why a change has not been made. We hope you will find the comments helpful and that you will be able to return your revised manuscript within 2 weeks. If you require longer than this, please contact journal staff: jp@physoc.org.

REVISION CHECKLIST:

We look forward to receiving your revised submission.

Yours sincerely,

Vaughan Macefield
Senior Editor
The Journal of Physiology

REQUIRED ITEMS

- The reference list must be in alphabetical order, rather than numbered, to comply with our Journal format.

EDITOR COMMENTS

Reviewing Editor:

The authors have adequately addressed the comments from the reviewer and reviewing editor. One minor point is that the references do not currently follow the Journal of Physiology's style and should be changed.

Senior Editor:

Thank you for submitting your revised manuscript to The Journal of Physiology. While the reviewer is satisfied by your amendments, as noted by the Reviewing Editor, please format the references according to Journal style: references should be cited in the text as (Surname et al., Year), not numbered.

REFEREE COMMENTS

Referee #1:

The authors have been very responsive and provided clarity regarding Trpv1+ fibers versus Trpv1 channels. No further comments.

END OF COMMENTS

Senior Editor:

Thank you for submitting your revised manuscript to The Journal of Physiology. While the reviewer is satisfied by your amendments, as noted by the Reviewing Editor, please format the references according to Journal style: references should be cited in the text as (Surname et al., Year), not numbered.

Response:

Thank you for your comments regarding our manuscript. We have revised the format of the references to be in accord with Journal of Physiology format.

Dear Dr Osborn,

Re: JP-P-2025-290333R2 ""When the brain listens to the kidney, is it TRP'n or what?"" by John W Osborn, Louise C. Evans, Christopher T Banek, Lucy Vulchanova, and Alex Dayton

Thank you for submitting your manuscript to The Journal of Physiology. It has been assessed by a Reviewing Editor and we are pleased to tell you that it is acceptable for publication following satisfactory minor revision.

The review comments are copied at the end of this email.

Please address all the points raised and incorporate all requested revisions or explain in your Response to Referees why a change has not been made. We hope you will find the comments helpful and that you will be able to return your revised manuscript within 2 weeks. If you require longer than this, please contact journal staff: jp@physoc.org.

REVISION CHECKLIST:

We look forward to receiving your revised submission.

Yours sincerely,

Vaughan Macefield
Senior Editor
The Journal of Physiology

EDITOR COMMENTS

Thank you for reformatting the references. Before final acceptance could I trouble you to resubmit after leaving a space before each reference citation, e.g. "using the TRPV1-/- rat(Sullivan et al., 2025)" should be "using the TRPV1-/- rat (Sullivan et al., 2025)"

Also "increase in renal pelvic pressure(Sullivan et al., 2025)" should be "increase in renal pelvic pressure (Sullivan et al., 2025)"

And: "estimated by Stocker and Sullivan(Stocker & Sullivan, 2023)" should simply be "estimated by Stocker and Sullivan (2023)"

END OF COMMENTS

EDITOR COMMENTS

Thank you for reformatting the references. Before final acceptance could I trouble you to resubmit after leaving a space before each reference citation, e.g. "using the TRPV1-/- rat(Sullivan et al., 2025)" should be "using the TRPV1-/- rat (Sullivan et al., 2025)"

Also "increase in renal pelvic pressure(Sullivan et al., 2025)" should be "increase in renal pelvic pressure (Sullivan et al., 2025)"

And: "estimated by Stocker and Sullivan(Stocker & Sullivan, 2023)" should simply be "estimated by Stocker and Sullivan (2023)"

RESPONSE:

We have made the requested changes regarding spacing before each reference citation.

Dear Professor Osborn,

Re: JP-P-2025-290333R3 ""When the brain listens to the kidney, is it TRP'n or what?"" by John W Osborn, Louise C. Evans, Christopher T Banek, Lucy Vulchanova, and Alex Dayton

We are pleased to tell you that your paper has been accepted for publication in The Journal of Physiology.

Please see below for a minor comment that can be amended at proof stage.

Please note that Perspective articles are not typically covered by institutional open access agreements with our publisher, Wiley. Wiley do not offer article processing charge (APC) discounts for smaller article types in hybrid subscription journals, meaning that if you wish for your Perspective to be published Open Access, you will have to pay the full APC. As such, we recommend authors publish Perspectives 'behind the paywall', where they will become freely accessible after a 12-month embargo (i.e. please select the NON open access option via Wiley Author services during proofing).

Should you wish to pay for Open Access, you will be able to place an order by logging into Wiley Author services.

Yours sincerely,

Vaughan Macefield
Senior Editor
The Journal of Physiology

IMPORTANT POINTS TO NOTE FOLLOWING ACCEPTANCE OF YOUR PAPER:

- **IMPORTANT NOTICE ABOUT OPEN ACCESS:** To assist authors whose funding agencies mandate immediate public access to published research findings, The Journal of Physiology allows authors to pay an Open Access (OA) fee to have their papers made freely available immediately on publication.

- You can help your research get the attention it deserves! Check out Wiley's free Promotion Guide for best-practice recommendations for promoting your work at: www.wileyauthors.com/eeo/guide. You can learn more about Wiley Editing Services which offers professional video, design, and writing services to create shareable video abstracts, infographics, conference posters, lay summaries, and research news stories for your research at: www.wileyauthors.com/eeo/promotion.

- If you would like to receive our 'Research Roundup', a monthly newsletter highlighting the cutting-edge research published in The Physiological Society's family of journals (The Journal of Physiology, Experimental Physiology, Physiological Reports, The Journal of Nutritional Physiology and The Journal of Precision Medicine: Health and Disease), please click this link, fill

in your name and email address and select 'Research Roundup':
<https://www.physoc.org/journals-and-media/membernews>

EDITOR COMMENTS

Dear John,

Thank you for your excellent contribution and the snappy title. I am pleased to report that it is now acceptable for publication in The Journal of Physiology.

At the proof stage you will need to correct the spelling of "Stocker and Sullivan (Stocker & Sullivan, 2023)" and simply cite the reference as "Stocker and Sullivan (2023)"